# The Astonishing Large Family of HSP40/DnaJ Proteins Existing in *Leishmania*

**DOI:** 10.3390/genes13050742

**Published:** 2022-04-23

**Authors:** Jose Carlos Solana, Lorena Bernardo, Javier Moreno, Begoña Aguado, Jose M. Requena

**Affiliations:** 1Centro de Biología Molecular Severo Ochoa (CSIC-UAM), Departamento de Biología Molecular, Instituto Universitario de Biología Molecular (IUBM), Universidad Autónoma de Madrid, 28049 Madrid, Spain; jcsolana@cbm.csic.es; 2Centro de Investigación Biomédica en Red de Enfermedades Infecciosas (CIBERINFEC), Instituto de Salud Carlos III, 28029 Madrid, Spain; lorena.bernardo@isciii.es (L.B.); javier.moreno@isciii.es (J.M.); 3WHO Collaborating Centre for Leishmaniasis, Centro Nacional de Microbiología, Instituto de Salud Carlos III, Majadahonda, 28220 Madrid, Spain; 4Centro de Biología Molecular Severo Ochoa (CSIC-UAM), Genomic and NGS Facility (GENGS), 28049 Madrid, Spain; baguado@cbm.csic.es

**Keywords:** *Leishmania*, HSP40, J-domain protein (JDP), molecular chaperones

## Abstract

Abrupt environmental changes are faced by *Leishmania* parasites during transmission from a poikilothermic insect vector to a warm-blooded host. Adaptation to harsh environmental conditions, such as nutrient deprivation, hypoxia, oxidative stress and heat shock needs to be accomplished by rapid reconfiguration of gene expression and remodeling of protein interaction networks. Chaperones play a central role in the maintenance of cellular homeostasis, and they are responsible for crucial tasks such as correct folding of nascent proteins, protein translocation across different subcellular compartments, avoiding protein aggregates and elimination of damaged proteins. Nearly one percent of the gene content in the *Leishmania* genome corresponds to members of the HSP40 family, a group of proteins that assist HSP70s in a variety of cellular functions. Despite their expected relevance in the parasite biology and infectivity, little is known about their functions or partnership with the different *Leishmania* HSP70s. Here, we summarize the structural features of the 72 HSP40 proteins encoded in the *Leishmania infantum* genome and their classification into four categories. A review of proteomic data, together with orthology analyses, allow us to postulate cellular locations and possible functional roles for some of them. A detailed study of the members of this family would provide valuable information and opportunities for drug discovery and improvement of current treatments against leishmaniasis.

## 1. Introduction

Life is the result of an orchestrated interplay between the biomolecules constituting cells and environmental signals, which are of either biological, physical or chemical nature. Inside the cells, proteins are continuously being synthesized, and they have to reach a proper folding to perform their biological functions, but these processes occur in a ‘crowded marketplace’. In this context, it is not surprising that molecular chaperones were among the first inhabitants of living cells. These proteins, many of them also known as heat shock proteins (HSPs) because they are involved in protecting cells from the effects derived of anomalously high temperatures, assist the folding and assembly of most de novo synthetized polypeptides. Moreover, molecular chaperones are involved in cell-signaling cascades through modulating interactions between different biomolecules. There are several families of evolutionarily conserved chaperones, but central is the HSP70 chaperone, which is considered the master player in protein homeostasis [1]. Nevertheless, the functional versatility of Hsp70, its precise localization within the cell and the specificity of substrate binding are substantially dependent on its interaction with a family of co-chaperones named either HSP40s, JDPs (DnaJ-domain-containing proteins) or J-proteins [2]. This review presents an overview of the structural features and functional roles of the HSP40 family members in *Leishmania*, a parasite possessing one of the largest compendiums of different HSP40s.

## 2. The Stressful Life of *Leishmania* Parasites

*Leishmania* parasites are the causative agent of the leishmaniases, a group of diverse diseases ranging in severity from a spontaneously healing skin ulcer to disfiguring mucocutaneous lesions or visceral disease, the latter often fatal if untreated [3]. More than 12 million people suffer some form of the disease in tropical and subtropical regions of the world, including Central and South America, the Indian subcontinent, the Middle East, Central Asia, East Africa and the Mediterranean basin [4].

This protist is transmitted to mammalians by insects; to complete its life cycle, therefore, the parasite requires adaptation to two different hosts. The promastigote stage (easily recognized by its flagellum) lives in the gut of phlebotomines (a.k.a., sand flies), mainly of the genera *Phlebotomus, Lutzomyia* and *Psychodopygus*. When the sand fly vector bites to feed on the mammalian host, promastigotes are inoculated into the dermis. Then, promastigotes are phagocytized by macrophages, but the parasites are able to survive inside the mature phagolysosome compartment, where they differentiate to the non-motile amastigote stage. Therefore, along its life cycle, *Leishmania* faces dramatic changes in the environmental conditions, including temperature upshifts, acidic pH, oxidative stress and nutrient deprivation [5]. Moreover, these changes are cyclical and reversible, when the parasite is transmitted back from the mammalian to the insect vector. Most of the environmental changes faced by the parasite might have a deleterious impact on structure and function of many *Leishmania* proteins. In order to counteract these effects, cells have evolved the stress response, in which molecular chaperones are central components. As many molecular chaperones were first identified as being induced by heat shock, they are also known as heat shock proteins (HSPs) or in general stress proteins [6]. In *Leishmania* parasites, because of their stressful life cycle, a robust and versatile chaperone system was developed in order not only to act as cytoprotector against different stresses but also to tune stage differentiation and virulence development [7].

## 3. The HSP70/HSP40 Chaperone System

Protein quality control is paramount for all molecular processes mediated by protein interactions that occur along the cell life. Together with protease systems and cellular mechanisms such as autophagy and lysosomal degradation, chaperones ensure that proteins are correctly folded and functional at the right place and time [8]. Molecular chaperones are nanomachines specially engineered to interact with non-native conformations of proteins to avoid protein aggregation and assist protein folding. Usually, they bind to hydrophobic residues that are transiently exposed during initial folding but also present in damaged or denatured proteins; in their absence, protein aggregation may occur. The main classes of molecular chaperones are grouped in families, named according to the molecular weights of their prototypical members: small HPS (sHSPs), HSP40, HSP60, HSP70, HSP90 and HSP100 [8,9].

Among molecular chaperones, members of the HSP70 family are central hubs of many cellular processes [10]. Because of this, it is not surprising that prototypical HSP70 (named DnaK in bacteria) is the most conserved protein present in all organisms [11]. Moreover, HSP70 chaperones are key in protein quality control mechanisms. Thus, HSP70s interact with nascent polypeptides at the ribosome to assist de novo protein folding and with mis-folded or stress-denatured proteins; also, they are involved in the stabilization of partially unfolded proteins prior their translocation across membranes [12,13,14].

HSP70s have two functional domains, a substrate binding domain (SBD) and a nucleotide-binding domain (NBD). These domains work in a coordinate manner to accommodate the diverse functional roles played by HSP70s. Thus, the SBD moiety interacts with the target proteins, but this binding is transient, being modulated by an allosteric mechanism that involves cycles of ATP hydrolysis and ADP/ATP exchange in the NBD moiety (Figure 1). Briefly, in the ATP-bound state, the HSP70 presents low affinity and fast exchange rates for the polypeptide substrates, while in the ADP-bound state, HSP70 has high affinity and low exchange rates for substrates [10]. When ATP binds to the NBD, the entire HSP70 structure is affected, and the substrate-binding cleft of SBD opens to release the substrate (polypeptide). The subsequent ATP hydrolysis leads to a new conformational alteration (ADP-bound state) in the HSP70 structure that results in the trapping and tight binding of the target protein [10]. Thus, binding and release alternating steps are important to stabilize non-native proteins and to promote their correct folding. Remarkably, HSP70s possess a weak intrinsic ATPase activity, but this is activated upon interaction with co-chaperones of the family HSP40/JDP. In fact, JDPs play a double role: they favor the binding of substrate proteins to HSP70 in its ATP-bound stage, but concomitantly a direct JDP-HSP70 interaction triggers the ATP hydrolysis and the transition to the ADP-bound state of HSP70, which has high affinity for the specific polypeptide brought over by a particular JDP. Another crucial player is the nucleotide exchange factor (NEF), which induces ADP dissociation and binding of a new ATP molecule, modifying HSP70 structure to the low-affinity state and consequently leading to substrate release [15].

## 4. J-Domain Proteins

HSP40s, also known as J-domain proteins (JDP), DNAJ-like or J proteins, are essential partners for HSP70 chaperones [16]. These proteins are grouped because of the presence of a J-domain, whose prototype is that defined in *Escherichia coli* DnaJ protein [17]. JDPs can bind to substrate polypeptides by themselves, and their J-domain promotes ATP hydrolysis by the HSP70 protein, favoring the binding of polypeptides by the HSP70 [18]. Remarkably, but not surprisingly as they are responsible for substrate specificity, HSP40s, in a cell or into cellular compartments, outnumber HSP70 family members [19]. Thus, 6 HSP40s have been found in *E. coli*, 22 in *Saccharomyces cerevisiae*, 41 in humans [20], 49 in *Plasmodium falciparum* [21] and 69 in *L. major* [7]. In contrast, only three distinct DnaK genes exist in *E. coli*; six distinct HSP70s are present in *P. falciparum*, nine in *Leishmania* and ten in humans [7].

As mentioned above, the prototypical and founding member of this superfamily is the *E. coli* DnaJ protein [17], whose structure and functions have been elucidated in great detail [22]. DnaJ contains four structural domains: an N-terminal J-domain, followed by a Gly/Phe (G-F)-rich domain, a Zn^2+^-finger domain and a less-conserved C-terminal domain. The J-domain region is comprised of approximately 70 amino acids that fold into four α-helices (I–IV). The existence of a highly conserved His-Pro-Asp (HPD) tripeptide motif in the loop region between helices II and III is another structural feature of the J-domain; this motif is essential for the stimulation of HSP70 ATP hydrolysis [23]. It is believed that J-domain only interacts with the ATP-bound HSP70 conformation at the interface between NBD and SBD moieties. Then, the HDP motif contacts key residues of the HSP70 ATP catalytic site, remodeling the NBD lobes to orientate the catalytic residues to a position optimal for ATP hydrolysis. Furthermore, the J-domain interacts with residues of the HSP70 SBD, promoting high affinity for the HSP70 ADP-bound state and efficient trapping of substrates [10,22]. Moreover, many J-proteins directly bind substrates, favoring the specific interactions of HSP70 with particular polypeptides and linking in turn the HSP70 functions to particular cellular processes [2]. Apart from the common J-domain, which is the distinctive feature of HSP40s, this class of proteins show a large structural divergence among them. Different HSP40s interact with a particular member of the HSP70 family; hence, HSP40s are contributing to the multi-functionality of the HSP70 machinery by specifying the target substrates and cellular processes in which the HSP70 chaperone activity is requested [20,24].

Based on the similarity to the domain architecture of the DnaJ, HSP40s have been grouped into four classes (Figure 2). Type I proteins share the four characteristic domains of DnaJ: the N-terminal domain constituted by the J-domain; a glycine-phenylalanine (G-F)-rich linker segment; two β-sandwich C-terminal domains, which contain four repeats of the CxxCxGxG type (zinc finger-like region); and a dimerization domain involved in binding to the client proteins. Examples of proteins belonging to this class are the yeast Ydj1 and the human DnaJA1-4. Type II proteins share the J-domain, the G-F-rich linker and the C-terminal substrate binding domain but lack the zinc-finger domain; examples are the yeast Sis1 and the human DnaJB-1, -4 and -5 [24]. The G-F-rich region is also involved in determining the specificity of HSP40s for target proteins [25]. Additionally, in class II JDPs, the G-F-rich region would be involved in an autoinhibitory mechanism, in which the G-F region initially blocks J-domain binding to HSP70 [15]. Type III proteins are heterogeneous in sequence and share only the J-domain with DnaJ; more often, they contain domains involved in specifying target interaction or sub-cellular localization [20]. As mentioned above, within the J-domain, there exists the highly conserved HPD motif; however, for some J-domain containing proteins, a strict HPD sequence is not found. To denote this feature, Louw and coworkers proposed a fourth group (type IV) of HSP40s to include those proteins lacking the HPD sequence in their J-domains [26]. Unlike type I and type II HSP40s, in which the J-domain always has an N-terminal location, in type III proteins the J-domain can be in any position along their sequence. It has been suggested that type I and type II HSP40s form complexes (dimers or tetramers) and are able to interact promiscuously with nascent polypeptides; also, they recognize mis-folded or aggregated proteins and cooperate with HSP70s in protein disaggregation [10,27]. In contrast, type III HSP40s would have evolved to specifically interact with a limited number of HSP70 substrates or alternatively acting directly as a bait to locate an HSP70 to particular cellular places [10,20]. Regarding type IV HSP40s, some authors have questioned whether they must be considered either members of the JDP family or only JDP-like proteins [28]. Nevertheless, they should be considered to understand the evolutionary history of the family and its functional diversification. Moreover, for some HSP40s, the maintenance of their co-chaperone functions in the absence of a canonical J-domain has been reported [29].

## 5. Appraisal and Updating of the Compendium of HSP40s in *L. infantum*

In a previous work, we found 69 different HSP40s to be encoded in the *L. major* genome [7]. At that time (2015), the *L. major* genome was the best assembled one for the genus *Leishmania*; however, in 2017, an improved genome assembly was attained by the combination of second- and third-generation sequencing methodologies for the *L. infantum* genome [30]. Hence, in this review, we analyzed the *L. infantum* genome for genes encoding J-domain containing proteins. All the 69 previously identified proteins for the HSP40 family in *L. major* were found to be also present in the *L. infantum* genome. Moreover, three new members were uncovered, amounting to a total of 72 different JDPs (Figure 3 and Table 1). Following the nomenclature proposed by Folgueira and Requena [31], they were named J75 (JDP75), J76 (JDP76) and J77 (JDP77). Thus, *Leishmania* possesses one of the largest HSP40 families among the organisms in which this family has been studied. To our knowledge, a larger HSP40 family has only been found in pepper, which contains 76 annotated HSP40 genes in its genome [32]. According to the presence/absence of the prototypical DnaJ-domains (Figure 2), the 72 *Leishmania* JDPs could be grouped into the four established classes (Table 1). Thus, eight *L. infantum* HSP40 proteins belong to the type I category, since they contain all typical domains found in the prototypical DnaJ molecule; these are J2, J3, J4, J27, J32, J45, J46 and J50. Another 18 HSP40s belong to the type II, as they have Gly/Phe-rich region close to the J-domain but lack a zinc-binding domain. The largest category (type III) is that formed by 38 proteins containing only the J-domain. Of note, the protein J30 (gene LINF_070013700), as currently annotated at TriTrypDB, lacks the J-domain; however, its ortholog in *Trypanosoma brucei* (Tb927.8.1010) contains the J-domain at the N-terminal moiety. Thus, we analyzed the LINF_070013700 transcript sequence and found that the gene was mis-annotated, and the coding sequence can be extended 783 nucleotides at its 5′-end. Thus, the new predicted protein would be 261 amino acids longer and then would contain the complete J-domain [33]. The *L. infantum* J10 protein (gene LINF_170010900) was found to be N-terminal truncated related to its *L. major* ortholog, and, as a consequence, the J-domain is incomplete, although the HPD motif is present; in this case, a possible mis-annotation of the coding sequence was not evidenced [34]. Within the type IV HSP40 group, six proteins were found to be encoded in the *L. infantum* genome: J31, J47, J66, J68, J75 and J77. Of them, J47, J66 and J75 have also the Zn-finger domain (Table 1).

Among the members of the *Leishmania* HSP40 family, as observed in other organisms, there is little sequence conservation beyond the characteristic J-domain. A phylogenetic analysis was conducted with the 72 JDPs to determine possible evolutionary relationships (Figure 3). Only type I HSP40s (except J32) grouped in the same branch of the tree, but the small bootstrap values supporting most of the branches did not allow for definition of clear evolutionary relationships among them. Exceptions are the pairs J45 and J46 (bootstrap 99%) and J6 and J7 (96%), proteins that probably resulted from a recent duplication of an ancestral gene. Proteins of the type II, III and IV show marked divergences in their structure and sequence; even the J-domain (typically found at the N-terminal region in DnaJ and type I-HSP40s) is located in the middle of the sequence or even at the C-terminal region in the proteins J20, J21, J25, J28, J29, J35, J41, J42, J51, J52, J53, J64, J65, J67 and J75 (Table 1). This atypical location of J-domain has been also reported for HSP40s in other organisms [37]. A remarkable feature is that *Leishmania* J14 contains two J-domains.

*Leishmania* protists belong to the Euglenozoa phylum, which represent one of the earliest extant branches of the eukaryotic lineage [38]. Thus, the enormous evolutionary distance separating *Leishmania* from the model eukaryotes makes it difficult to establish straightforward orthologous relationships between *Leishmania* proteins and the human or yeast ones; in fact, in databases, around half of the *Leishmania* genes are annotated as coding for proteins of unknown functions because no clear orthologous proteins exist in model eukaryotes. Nevertheless, based on the hypothesis that the identification of potential orthologues in the well-characterized human and/or yeast proteomes would give clues about the functions of the *Leishmania* proteins, we performed a detailed search of sequence conservation with the 72 *L. infantum* HSP40s against protein databases found in the servers PantherDB (http://www.pantherdb.org/, accessed on 22 April 2022), Expasy (https://prosite.expasy.org/, accessed on 22 April 2022), InterPro (https://www.ebi.ac.uk/interpro/, accessed on 22 April 2022) and UniProt (https://www.uniprot.org/blast/, accessed on 22 April 2022). As a result, a possible orthologous relationship among the *Leishmania* type I J2, J3 and J4 (these three proteins may share an evolutionary origin, see Figure 3) and the members DnaJA-1, -2 and -4 of the human HSP40 family was deduced. Similarly, J2 may be ortholog to *S. cerevisiae* mas5 protein (YDJ1 gene), which is involved in mitochondrial protein import [39]. Also, *Leishmania* type I-JDPs J45, J46 and J50 have substantial sequence conservation with the human DnaJA2 and *S. cerevisiae* SCJ1 proteins, which are located at the endoplasmic reticulum in those organisms [40,41].

Within the group of type IV HSP40s, J66 has a probable evolutionary relationship with *L.*
*infantum* type I-JDPs and particularly with J2 (Figure 3). Thus, J2 and J66 may be considered paralogs, and in turn both would be orthologs to human DnaJA2. Another type IV-JDP, J47, shares sequence conservation with the type I-JDP J27 (Figure 3), and both are possible orthologs to human DnaJA3 and *S. cerevisiae* Mdj1 proteins, which were located at the mitochondrion [42,43].

J6 and J7, belonging to the type II group, might be orthologs to human DnaJB4, DnaJB5 and DnaJB1 HSP40s and to *S. cerevisiae* Sis1 protein, which is required for nuclear migration during mitosis and initiation of translation [44,45]. J10 (and to some extent J34) may be orthologue to Sec63, an essential HSP40 protein involved in post-translational translocation of proteins across the endoplasmic reticulum [46,47]. Another endoplasmic reticulum located proteins, human DnaJC3/ERdj6 and *S. cerevisiae* JEM1 [48,49], might be the orthologs of *Leishmania* J53. Moreover, we postulate that *Leishmania* J13 is ortholog of human DnaJC24, a protein involved in diphthamide biosynthesis, a post-translational modification of histidines that have been found in the translation elongation factor-2 [50]. A possible ortholog of J16 would be the well-known human zuotin/DnaJC2 protein. Zuotin is a component of the ribosome-associated complex involved in maintaining nascent polypeptides in a folding-competent state [51]. The type I-JDP J27 may be an ortholog of the human DnaJA3/Tid1 protein, a mitochondrial molecular chaperone [42,52]. A plausible mitochondrial location of *Leishmania* J31 is suggested on its sequence similarity with human DnaJC11 [53]. J36 might be ortholog of human DnaJC20/HscB and *S. cerevisiae* JAC1 proteins, which are involved in iron–sulfur cluster biosynthesis [54,55]. *Leishmania* J68 might be a component of the mitochondrial Tim translocase because of its structural and sequence similarities with human DnaJC15 and *S. cerevisisae* Pam18 proteins [56]. Human DnaJC21 and *S. cerevisiae* JJJ1 are involved in rRNA biogenesis [57], and they may be orthologs of *Leishmania* J32. A histone chaperone function may be suggested for *Leishmania* J33 based on its sequence similarity with human DnaJC9 and *Schizosaccharomyces pombe* C1071.09c proteins [58]. J59 is a really long protein (2451 amino acids) that might be orthologue of the human DnaJC13/RME-8 protein, which is involved in endosome organization and regulation [59]. In addition to all possible orthologs mentioned above, another several *Leishmania* HSP40s share remarkable sequence identity with genes coding for HSP40s in fungi and plant species. Nevertheless, these are not commented here, as they remain still uncharacterized, and no information about their functional roles is available.

According to structural features and the putative orthologs identified (see above), some *Leishmania* HSP40s could be associated to distinct sub-cellular locations such as mitochondrion (J27, J31, J36, J47 and J68), endoplasmic reticulum (J10, J22, J34, J45, J46, J53, J66 and J72), flagellar pocket (J11, J51 and J54), endosomes (J59) and the nucleus (J14, J33 and J56) (Table 2). Of note, several *Leishmania* HSP40s (J2, J3, J6, J8, J27, J50 and J54) have been localized in the *L. donovani* glycosomes following a proteomic approach [60] (Table 2). Glycosomes are specialized peroxisomes, existing in *Leishmania* and other trypanosomatids, that contain key enzymes involved on the glycolytic pathway and purine salvage [61]. Moreover, at least 23 HSP40s in *Leishmania* may play functional roles in cellular membranes as they possess transmembrane domains, including members from group I (J45, J46), group II (J5, J19, J22, J28, J44, J53 and J54) and group III (Table 1). In these locations, putative functions for these *Leishmania* JDPs may be providing co-translational chaperone assistance, transport across organelle membranes and unfolding/refolding after transportation from one cellular compartment to another. For example, J22 is a possible membrane protein related to endoplasmic reticulum (ER) membrane-located *S. pombe* pi041 and human DnaJB12 proteins that are involved in ER-associated degradation of misfolded proteins [62].

In addition, TPR (tetratrico-peptide repeat) domains have been found in eight JDP proteins: J51, J52, J53, J67 and J76 contain two or more TPR domains each, whereas a sole TPR domain is present in J42, J65 and J76 (Table 1). TPR domain has been found to be a docking site, interacting with the EEVD motif present at the C-termini of some members of the HSP70 family and in the C-terminus of the HSP83/90 protein [73,74].

Despite the outstanding number of HSP40s existing in *Leishmania*, none of these proteins have been biochemically characterized to date, and consequently, nothing is known about the potential HSP70-HSP40 partnerships and their role in the *Leishmania* life cycle. However, evidence of their existence and levels of stage specific expression are available now from transcriptomics and proteomics data. Thus, out of the 72 putative HSP40s identified in *Leishmania*, 37 have been experimentally reported in proteomic studies of the *Leishmania* promastigote [62,66,72] and/or amastigote stages [67,70]. Moreover, some of them have been found in the proteomes derived from specific sub-cellular fractions of the parasite such as the glycosome [60] and the flagellum [72]. A list of the experimentally detected HSP40s in *Leishmania* proteomes is included in Appendix A. For example, J2 protein has been experimentally reported in *L. infantum* and *L. donovani* promastigotes and amastigotes [60,63,66], in extracellular vesicles of *L. infantum* [68] and in the *L. braziliensis* secreted proteome [75]. Remarkably, J2 has been found to be phosphorylated on Serine-89, and the phosphorylation ratio of this residue increased by 3.5-fold after 2.5 h of promastigote-to-amastigote differentiation, reaching an increase in phosphorylation up to 22-fold in full differentiated amastigotes [76]. Such a dramatic increase in phosphorylation suggests a relevant role for this protein in the differentiation process from promastigote to amastigote stage. Phosphorylation on serine residues is the most common post-transcriptional modification in HSP40s (Appendix A) and may be related to a general regulatory mechanism induced during the stress response [77]. Other relevant protein modifications found in HSP40s are acetylation and methylation. Particularly relevant may be the N-terminal acetylation on threonine-39 of J7 [78], because this modification would be changing its chemical properties and having marked biological consequences on protein function and cellular localization [64]. In addition, methylation on glutamic acid-134 in J6 and J50 proteins would increase the hydrophobicity of this protein [79].

## 6. Concluding Remarks and Future Work

The HSP40 proteins play essential roles in cellular metabolism by regulating interactions between HSP70 chaperones and their client proteins. We identified 72 HSP40 coding genes in the *Leishmania* genome, making up an exceptionally large protein family for this parasite compared to other eukaryotes. However, unlike other insect-transmitted parasites such as *Plasmodium* [80], the crucial role of HSP40s in the survival and pathogenesis of *Leishmania* parasites remains unexplored. Future studies should address the molecular characterization of the HSP40 family members and the identification of their HSP70 counterparts. Different HSP40s can bind and deliver client proteins to a single HSP70 and establish unique HSP70–HSP40 pairs with specific activities at distinct cellular locations. Thus, studies aimed to analyze the co-localization of particular HSP70 family members and HSP40s in sub-cellular compartments would be a suitable strategy to establish HSP70–HSP40 partnerships. For this purpose, high throughput tagging based on CRISPR/Cas9 tools would be a strategy for those localization studies [81]. Another appealing approach to define specific HSP40 location would be the use of protein phase separation methods; in fact, many HSP40s have been found as components of membraneless organelles such as nucleolus or stress granules [63]. Also, CRISPR-Cas9 genome editing methods would be useful for analyzing essential roles played by this family of proteins, following the strategies used to screen proteins involved in *Leishmania* flagellar architecture and function [72] or to define the essential kinases comprising the *Leishmania* kinome [82]. In addition, as HSP40s regulate the activity of the HSP70 chaperone system, the analysis of changes in gene expression of *HSP40* genes along the *Leishmania* life cycle, which includes a drastic morphological differentiation and adaptations to stressful environments, would inform on the relevance role played by every HSP40 at particular points of the life cycle. These studies also might be useful to detect changes in the parasite fitness related to drug resistance. Thus, for instance, it has been reported that transcription of the *J57* gene is upregulated in an *L. donovani* line resistant to liposomal amphotericin B (AmBisome) drug, the most effective anti-*Leishmania* treatment [83]. In summary, we are just starting the fascinating road to comprehend the roles that this group of proteins are playing in the differentiation, survival and pathogenesis of *Leishmania* parasites.

## Figures and Tables

**Figure 1 genes-13-00742-f001:**
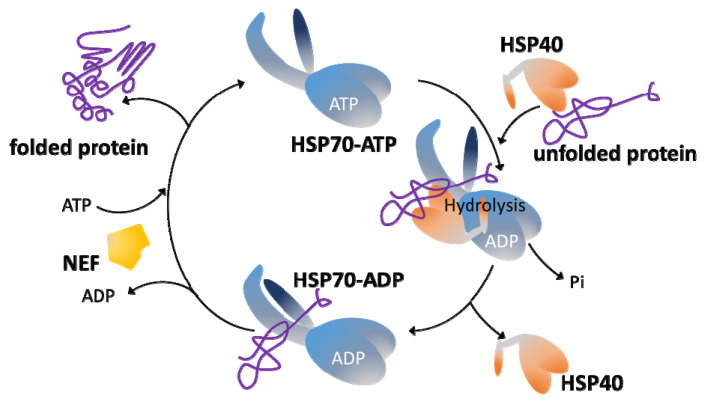
Overview of the HSP70/HSP40 chaperone system. An unfolded protein (the substrate) binds to an HSP40 member, and both form a complex with the HSP70 chaperone in its ATP-state, which has low affinity for polypeptides. The HSP40–HSP70 interaction triggers ATP hydrolysis and promotes a conformational change in the HSP70 (ADP-bound state) that results in a structure with high affinity for the protein substrate, which is bound into the substrate binding domain (SBD) of HSP70. Following substrate transfer, HSP40 leaves the complex, and the nucleotide exchange factor (NEF) is recruited to the HSP70–polypeptide complex, stimulating the ADP-by-ATP exchange. ATP binding induces both release of NEF and the folded polypeptide and leaves HSP70 ready for a new cycle.

**Figure 2 genes-13-00742-f002:**
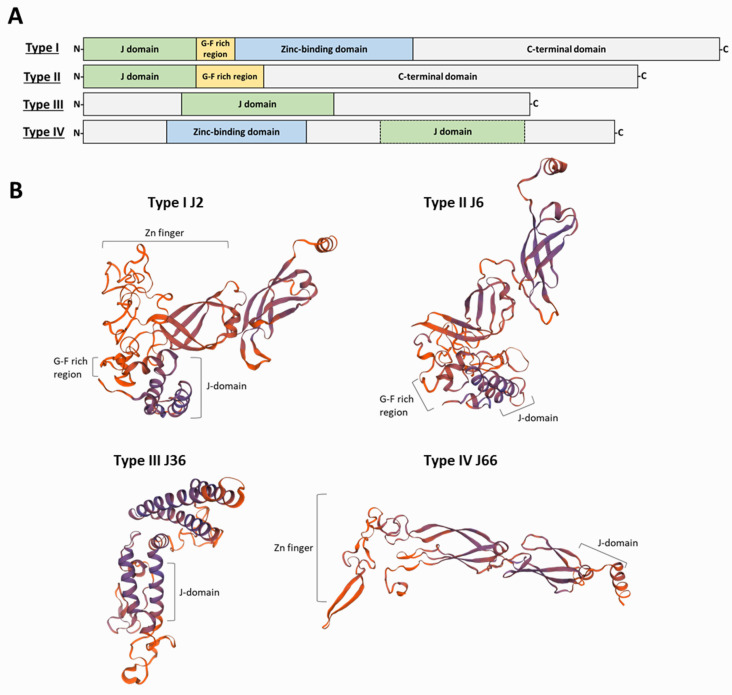
Classification of HSP40s into four types and representative *L. infantum* proteins for each type. (**A**) Type I proteins have in their structure the four typical domains, starting with a J-domain in the N-terminal end and a short G-F-rich region, followed by a zinc-binding domain that ends in the substrate-binding C-terminal domain. Type II proteins share the J-domain and the G-F-rich linker but lack the zinc-finger domain. Type III proteins show remarkable structure divergence and only share the J-domain, which is frequently located in the middle of the sequence. Type IV HSP40s include those proteins lacking the highly conserved HPD motif in their J-domains; the zinc-binding domain is absent in some type IV proteins. (**B**) Protein structure of representative members of the HSP40 family in *L. infantum.* The J-domain consists of four α-helices. The G-F-rich linker is followed by two β-sandwich C-terminal domains which contain four repeats of the CxxCxGxG motif (zinc finger region) and a dimerization domain that is involved in binding to client polypeptides.

**Figure 3 genes-13-00742-f003:**
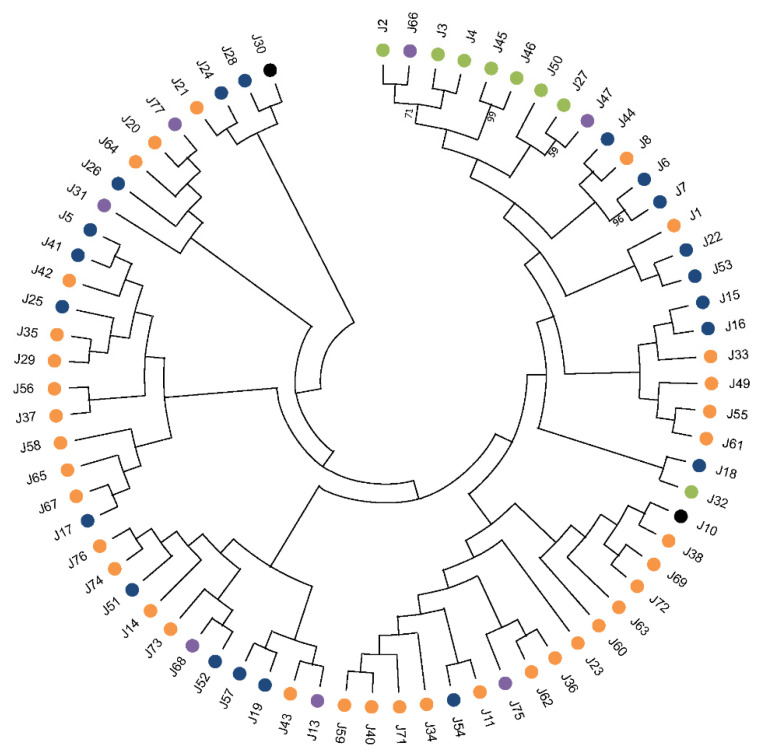
Phylogenetic relationships among *L. infantum* HSP40 proteins. The evolutionary history was inferred by using the Maximum Likelihood method and JTT matrix-based model [35]. The tree with the highest log likelihood (-96100,93) is shown. Initial tree(s) for the heuristic search were obtained automatically by applying Neighbor-Join and BioNJ algorithms to a matrix of pairwise distances estimated using the JTT model, and then selecting the topology with superior log likelihood value. This analysis involved 72 amino acid sequences, and the tree was inferred from 1000 replicates. There were a total of 2470 positions in the final dataset. Bootstrap values lower than 50% are not indicated. Evolutionary analyses were conducted in MEGA11 [36]. Color codes denote the HSP40 classification: type I (green), type II (blue), type III (orange), type IV (violet), not applicable (black).

**Table 1 genes-13-00742-t001:** *L. infantum* HSP40 proteins.

Name	Gene ID	Size	J-Domain	G/F Rich	Zn-Finger	Type	Remarks ^a^
J1	LINF_320037800	329	14–80	-	-	III	-
J2	LINF_270032200	396	6–72	78–94	120–205	I	-
J3	LINF_210010300	453	6–72	83–99	125–210	I	-
J4	LINF_150019800	478	6–72	81–97	144–227	I	-
J5	LINF_360019000	364	86–180	161–183	-	II	Transmembrane domains (192–215, 229–252).
J6	LINF_360072700	345	4–70	124–140	-	II	-
J7	LINF_320025300	323	9–75	91–107	-	II	-
J8	LINF_240010000	794	42–108	-	-	III	Signal peptide (1–24).
J10	LINF_170010900	157	-	-	-	-	Possibly mis-annotated (see text). HPD motif (5–7). Transmembrane domains (76–99, 113–136).
J11	LINF_040012900	576	9–75	-	-	III	Transmembrane domains (90–113, 127–150, 162–185, 194–217, 294–317, 346–369, 422–442, 461–484).
J13	LINF_180020400	184	14–80	-		III	-
J14	LINF_080014500	326	21–79/157–223	-	-	III	Two J-domains.
J15	LINF_190005600	432	5–71	128–149	-	II	-
J16	LINF_200016400	653	140–206	228–240	-	II	SANT/Myb SANT/Myb domain (513–567).
J17	LINF_120015200	608	4–70	64–126	-	II	-
J18	LINF_270009200	378	73–139	-	-	III	Transmembrane domains (330–353, 359–377).
J19	LINF_340048600	266	24–85	94–142	-	II	Transmembrane domain (129–152).
J20	LINF_360011700	260	171–237	-	-	III	J-domain at C-terminus.
J21	LINF_260019100	536	405–471	-	-	III	J-domain at C-terminus. Transmembrane domain (491–514).
J22	LINF_360028100	286	21–87	84–139	-	II	Transmembrane domain (149–172).
J23	LINF_180008300	244	49–115	-	-	III	Transmembrane domain (185–208).
J24	LINF_300022800	740	27–93	135–179	-	II	-
J25	LINF_260017700	898	459–525	535–551	-	II	J-domain in the middle.
J26	LINF_170005500	262	69–134	-	-	III	Transmembrane domain (170–193).
J27	LINF_040014400	493	92–158	164–194	253–331	I	-
J28	LINF_260017000	652	282–348	342–396	-	II	J-domain in the middle. Transmembrane domains (12–35, 110–133, 139–162).
J29	LINF_240016000	435	371–434	-	-	III	J-domain at C-terminus. Transmembrane domain (268–291).
J30	LINF_070013700	304	-	-	-	-	Truncated. Lacking first 242 aa.
J31	LINF_260014300	843	36–102	-	-	IV	-
J32	LINF_250029000	377	8–74	81–102	322–346	I	-
J33	LINF_360054000	275	7–73	-	-	III	-
J34	LINF_350052100	491	134–200	-	-	III	Transmembrane domains (12–35, 112–132, 218–241).
J35	LINF_140006000	523	377–443	-	-	III	J-domain at C-terminus.
J36	LINF_250023500	278	61–156	-	-	III	-
J37	LINF_180019800	1121	5–71	-	-	III	-
J38	LINF_300029900	336	15–81	-	-	III	-
J40	LINF_100017600	275	61–156	-	-	III	-
J41	LINF_310010500	603	288–354	373–396	-	II	J-domain at the middle.
J42	LINF_180022200	580	518–577	-	-	III	J-domain at C-terminus. Tetratricopeptide (TPR)-like helical domain (135–272).
J43	LINF_350045900	386	4–70	-	-	III	-
J44	LINF_310039900	217	18–84	105–148	-	II	Transmembrane domains (121–144).
J45	LINF_320040500	400	57–123	121–150	176–259	I	Transmembrane domain (12–32).
J46	LINF_250017100	396	57–123	123–150	175–258	I	Transmembrane domains (8–31).
J47	LINF_200010900	545	68–134	-	257–335	IV	-
J49	LINF_300015900	423	70–136	-	-	III	Prokaryotic lipoprotein domain (1–33).
J50	LINF_350035100	478	47–113	119–159	185–270	I	-
J51	LINF_340029700	808	700–766	779–807	-	II	J-domain at C-terminus. TPR region (345–471, 572–677). TPR domain (345–378, 384–417, 610–643, 644–677).
J52	LINF_360010300	510	386–452	461–509	-	II	J-domain at C-terminus. TPR region (17–118, 254–359). TPR domain (17–50, 51–84, 208–241, 254–287, 326–359).
J53	LINF_140019700	574	433–499	525–555	-	II	J-domain at C-terminus. Transmembrane domain (20–43). TPR region (51–120, 223–290).
J54	LINF_330016300	581	3–69	-	-	II	Transmembrane domains (90–113, 129–152, 156–179, 198–221, 240–263, 269–292, 416–439, 459–482).
J55	LINF_280018500	470	9–75	-	-	III	-
J56	LINF_330036200	266	77–133	-	-	III	-
J57	LINF_290026500	396	9–67	144–193	-	II	-
J58	LINF_240018200	808	5–68	-	-	III	-
J59	LINF_300027500	2451	1384–1450	-	-	III	2 GYF domain 2 (1059–1109).
J60	LINF_090022000	413	42–108	-	-	III	Prokaryotic lipoprotein domain (1–28).
J61	LINF_080011700	296	3–69	-	-	III	-
J62	LINF_340005300	679	95–161	-	-	III	Transmembrane domains (66–89, 174–197, 209–232).
J63	LINF_320011200	316	42–108	-	-	III	Transmembrane domain (278–301).
J64	LINF_070013600	417	170–252	-	-	III	J-domain in the middle. Transmembrane domain (362–385).
J65	LINF_340046400	690	616–682	-	-	III	J-domain at C-terminus. TPR-like helical domain (449–560).
J66	LINF_220005800	331	185–275	-	52–139	IV	-
J67	LINF_360013200	850	733–843	-	-	III	J-domain at C-terminus. TPR region (232–333, 569–640). TPR domain (232–265, 569–602, 607–640).
J68	LINF_240025300	121	54–120	-	-	IV	-
J69	LINF_360059200	346	19–77	-	-	III	-
J71	LINF_240005500	439	50–105	-	-	III	Transmembrane domain (318–337).
J72	LINF_350036200	428	20–86	-	-	III	Transmembrane domain (147–170).
J73	LINF_260031000	488	31–84	-	-	III	-
J74	LINF_280025400	578	66–119	-	-	III	-
J75	LINF_350007400	368	216–282	-	37–60	IV	J-domain at C-terminus. C3H1-type zinc finger (37–60).
J76	LINF_140005800	384	60–123	-	-	III	TPR region (1–31).
J77	LINF_250010800	197	79–148	-	-	IV	Transmembrane domain (160–183).

^a^ Data included in this column were obtained by using PROSITE (https://prosite.expasy.org/, accessed on 22 April 2022) or InterPro (https://www.ebi.ac.uk/interpro/, accessed on 22 April 2022) tools and from searching in public databases: PantherDB (http://www.pantherdb.org/, accessed on 22 April 2022), TriTrypDB (https://tritrypdb.org/, accessed on 22 April 2022) and UniProt (https://www.uniprot.org/, accessed on 22 April 2022).

**Table 2 genes-13-00742-t002:** *Leishmania* HSP40 members with possible orthologs in human and/or yeast.

Name	Ortholog [References]	Suggested Cellular Location ^a^	Identity (%)
J2	DNAJA1, DNAJA4, mas5 [39,63]	Glycosome, nucleolus	44.70-44.02
J3	DNAJA2 [40]	Glycosome	42.35
J4	DNAJA1, DNAJA4 [39,63]	Nucleolus	29.76, 31.97
J6	SIS1, DNAJB1, DNAJB4, DNAJB5 [44,45,64]	Glycosome, nucleus	35.03-38.83
J7	DNAJB4, DNAJB5, DNAJB1 [44,45]	-	33.24-32.16
J8	-	Glycosome	-
J10	Sec63 [46,47]	Endoplasmic reticulum (ER), nucleus	3.69
J11	-	Ciliary pocket	-
J13	DNAJC24 [50]	Cytoskeleton	27.38
J14	DNAJC8, SPF31 [65]	Nucleus	40.35, 29.41
J16	DNAJC2, zuotin [51]	Ribosome-associated complex	27.22, 35.89
J22	pi041, C17A3.05c, DNAJB12 [62]	ER membrane	27.51-28.57
J27	DNAJA3 [42,43,52]	Mitochondrion, glycosome	33.77
J31	DNAJC11, SPCC63.03 [53]	Mitochondrion	32.20, 27.27
J32	DNAJC21, JJJ1 [57]	-	40.34, 31.46
J33	DNAJC9, C1071.09c [58]	Nucleus, histone-related function	29.92, 31.25
J34	Sec63 [46,47]	ER membrane	31.10
J36	DNAJC20/HscB, JAC1 [54,55]	Mitochondrion	26.56, 23.61
J45	DNAJA2, SCJ1 [40,41]	ER	31.69, 31.40
J46	DNAJA2, SCJ1 [40,41]	ER	36.36, 30.66
J47	DNAJA3, MDJ1 [42,43]	Mitochondrion	24.79, 27.42
J50	DNAJA2, SPJ1, SCJ1 [40,41]	Glycosome	37.57-30.65
J51	DNAJC7 [66]	Ciliary basal body	28.07
J52	DNAJC7 [66]	-	32.77
J53	DNAJC3 (ERdj6), JEM1 [48,49]	ER	28.71, 38.94
J54	-	Flagellum, Glycosome	-
J56	DNAJC8 [65]	Nucleus	36.17
J59	DNAJC13, RME-8 [59]	Endosome	34.63, 34.63
J60	JJJ2v	-	38.89
J66	DNAJA2 [40]	ER	43.46
J68	DNAJC15 (tim complex) [56]	Mitochondrion	29.41

^a^ Proposed cellular locations are based on the location determined for ortholog proteins and/or data-derived from published proteomic studies in *Leishmania* [60,67,68,69,70,71,72].

## Data Availability

The data supporting reported results can be found at Leish-ESP (http://leish-esp.cbm.uam.es/, accessed on 22 April 2022), TriTrypDB (https://tritrypdb.org/, accessed on 22 April 2022), UniProt (https://www.uniprot.org/, accessed on 22 April 2022) and Wikidata datasets (https://www.wikidata.org/, accessed on 22 April 2022).

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
