# Peer review of "The Astonishing Large Family of HSP40/DnaJ Proteins Existing in Leishmania"

_genes, 2022, doi:10.3390/genes13050742_

Round 1

Reviewer 1 Report

The submitted manuscript “The astonishing large family of HSP40/DnaJ proteins existing in Leishmania” by Solana et al. is a review describing the diversity, structural features, and potential functions of this family of proteins. Overall, this is a well written review of the current knowledge on a large family of proteins and of interest to the scientific community, especially researchers in the Leishmania field. Fairly little is known about these proteins and thus the thorough sequence analyses and review of existing literature is valuable.

It is remarkable how many HSP40/DnaJ proteins exist and that no functional studies have been performed. This presents a large gap of knowledge that is valuable to address. In the conclusion section of the manuscript, the authors remark on that no functional studies have been performed, but provide no suggestions on how to best do this. Ideas on how to study such a high number proteins would strengthen the future work aspect in the concluding remarks. For example, high throughput tagging for localization studies, CRISPR/Cas mediated deletion studies, might enable functional assays in large protein families.

I find the extensive search of orthologs utilizing several data bases (PantherDB, Expasy, and UniProt) impressive and the outcomes interesting. The authors do devote a substantial – and justified – discussion on these potential functions; yet, the data is presented mostly in supplemental tables. The four tables in the supplemental section add valuable information to the manuscript. Table 1 (main text) lists the entirety of J proteins in Leishmania and the “Remarks” column includes structural information (including transmembrane motifs). It would be easy to add putative functions and localization into this table instead of relegating this information to supplemental table.s Alternatively, with Table 1 listing all the structural features, a second table could list orthologs with associated functions and cellular localizations (instead of listing this in the supplemental tables).

The information of supplementary table S2 (J proteins that may be localized to the glycosome) is also present in S1. The difference seemed to come from the sources of the information. Nevertheless, this could be combined into one table. The information from S3 (transmembrane domains) is included in table 1 of the main text and could thus be eliminated.

Some of the proteins may have unique cellular localization (as suggested from orthologs and/or proteomics data), for example the glycosome or mitochondria. Are there any targeting motifs (like the archetypical SKL or AKL tripeptides indicating glycosomal localization) within the sequences?

The summary report of HSP40 proteins being found in the promastigote and/or amastigote stage would benefit from the distinction if any studies were done that demonstrated if certain proteins being present in one stage but not the other (stage specific expression). Table S4 lists proteins that were found in promastigote and/or amastigote and most were found in the promastigotes stage. I assume that some of the cited studies were only performed in promastigotes (thus, no information is available on expression in the amastigote stage).

Minor comments:

DnaK is mentioned (line 91) but not defined.

The authors state that HSP40s outnumber HSP70 family members. What is a typical difference or ratio in numbers?

The authors reference previous work identifying 69 HSP40 proteins and a more recent study in 2017 that searched the updated L. infantum data bases. Here, it would make sense to state the previous work took place in 2015 (list both years, not just the recent one).

Line 213: why is the pepper HSP40 family with 76 annotated proteins larger than the Leishmania one with 77 proteins?

There are 8 type I, 20 type II, 38 type III, and 7 type IV HSP40/J proteins. Is there any significance to this distribution?

Although the manuscript is generally well written, there are a few awkward or grammatical incorrect sentences:

Line 39: “These proteins, many of them also known as 39 heat shock proteins (HSPs) because their involving in protecting cells…” should be “…because they are involved…”

Line 55: “…the latter often resulting fatal if untreated” should be “…the latter often resulting fatal if untreated

Line 62: “When the sand fly vector bites for feeding to the mammalian host…” should say something like “When the sand fly vector bites to feed on the mammalian host”.

Line 75: ” …because their stressful life cycles…” should be “…because of their stressful life cycles…”

Line 90: “members of HSP70 family” should be “members of the HSP70 family”.

Line 109: “…steps are important to stabilize non-native proteins and promoting their correct folding.” should be “steps are important to stabilize non-native proteins and to promote their correct folding.”

Line 30: “…whose prototype is that defined in Escherichia coli DnaJ” should be “…whose prototype is that defined in the Escherichia coli DnaJ protein”.

Line 133: “Remarkably, but no surprisingly…” should be “Remarkably, but not surprisingly”.
Line 144: “It is belief that J-domain…” should be “It is believed that the J-domain…”

Author Response

The submitted manuscript “The astonishing large family of HSP40/DnaJ proteins existing in Leishmania” by Solana et al. is a review describing the diversity, structural features, and potential functions of this family of proteins. Overall, this is a well written review of the current knowledge on a large family of proteins and of interest to the scientific community, especially researchers in the Leishmania field. Fairly little is known about these proteins and thus the thorough sequence analyses and review of existing literature is valuable.

Thank you for your overall comment on our manuscript and for your helpful suggestions.

It is remarkable how many HSP40/DnaJ proteins exist and that no functional studies have been performed. This presents a large gap of knowledge that is valuable to address. In the conclusion section of the manuscript, the authors remark on that no functional studies have been performed, but provide no suggestions on how to best do this. Ideas on how to study such a high number proteins would strengthen the future work aspect in the concluding remarks. For example, high throughput tagging for localization studies, CRISPR/Cas mediated deletion studies, might enable functional assays in large protein families.

In agreement with the reviewer’s suggestion, we have included, within the concluding remarks, a paragraph indicating the usefulness of high throughput CRISPR/Cas-based approaches in future works aimed to analyze the localization and functional roles of this large family of proteins. These approaches have already been used in a few studies in Leishmania, and those studies have been quoted in the revised manuscript.

I find the extensive search of orthologs utilizing several data bases (PantherDB, Expasy, and UniProt) impressive and the outcomes interesting. The authors do devote a substantial – and justified – discussion on these potential functions; yet, the data is presented mostly in supplemental tables. The four tables in the supplemental section add valuable information to the manuscript. Table 1 (main text) lists the entirety of J proteins in Leishmania and the “Remarks” column includes structural information (including transmembrane motifs). It would be easy to add putative functions and localization into this table instead of relegating this information to supplemental table.s Alternatively, with Table 1 listing all the structural features, a second table could list orthologs with associated functions and cellular localizations (instead of listing this in the supplemental tables).

Following these comments and suggestions, we have moved most of the supplementary data to Table 1. Additionally, we have created a second table (Table 2), in which the putative orthologues (from humans and yeasts) are listed together with the associated functions and cellular localization.

The information of supplementary table S2 (J proteins that may be localized to the glycosome) is also present in S1. The difference seemed to come from the sources of the information. Nevertheless, this could be combined into one table. The information from S3 (transmembrane domains) is included in table 1 of the main text and could thus be eliminated.

Following the reviewer’s suggestions, the information in supplementary tables S1 and S2 (containing somewhat redundant data) has been incorporated into the main Table 1. Also,  Table S3 was eliminated, because, as noted by the reviewer, the information from table S3 was already in Table 1.

Some of the proteins may have unique cellular localization (as suggested from orthologs and/or proteomics data), for example the glycosome or mitochondria. Are there any targeting motifs (like the archetypical SKL or AKL tripeptides indicating glycosomal localization) within the sequences?

This is also an interesting point. We have searched for archetypical peroxisomal targeting signals (PTS), either a PTS1 at their extreme carboxyl terminus, consisting of just three amino acids – SKL – or a conservative variant thereof, or a PTS2 motif (located close to the N-terminus, consensus sequence [RK]-[LVI]-x5-[HQ]-[LA]. None of the Leishmania HSP40 proteins has one of these PTS. In agreement, no HSP40s are among the L. major 191 potential PTS1-containing proteins and 68 potential PTS2-containing proteins identified by Opperdoes and Szikora after meaning the L. major genome [2006. In silico prediction of the glycosomal enzymes of Leishmania major and trypanosomes. Mol Biochem Parasitol 147, 193–206)].

            We can add this information if the Reviewer and/or Editor agree.

            Regarding the existence of a putative peptide signal for protein traffic into mitochondria, we are not aware that such sequences have been defined in Leishmania mitochondrial proteins.

The summary report of HSP40 proteins being found in the promastigote and/or amastigote stage would benefit from the distinction if any studies were done that demonstrated if certain proteins being present in one stage but not the other (stage specific expression). Table S4 lists proteins that were found in promastigote and/or amastigote and most were found in the promastigotes stage. I assume that some of the cited studies were only performed in promastigotes (thus, no information is available on expression in the amastigote stage).

In our opinion, none of the proteomics studies performed to date was designed to analyze stage-specific expression (promastigotes vs amastigotes). Hence, the presence/absence of a given protein cannot be considered as proof of stage specificity, as they may be representing random events of peptide detection (mainly if the number of identified peptides is low). There is an outstanding study, carried out by Dr Zilberstein’s group, in which a quantitative-proteomic study along the differentiation from promastigotes to axenic amastigotes was performed. Nevertheless, given the controversy existing whether or not axenic amastigotes are similar to tissue-derived amastigotes, we decided to not incorporate those data in Table S4.

Minor comments:

DnaK is mentioned (line 91) but not defined.

DnaK protein is the name used for the HSP70 ortholog in bacteria. This has been indicated in the revised manuscript.

The authors state that HSP40s outnumber HSP70 family members. What is a typical difference or ratio in numbers?

In response to this point, we have added the following sentence: In contrast, only three distinct DnaK genes exist in E. coli, six distinct HSP70s are present in P. falciparum, nine in Leishmania and ten in humans.

The authors reference previous work identifying 69 HSP40 proteins and a more recent study in 2017 that searched the updated L. infantum data bases. Here, it would make sense to state the previous work took place in 2015 (list both years, not just the recent one).

Following the reviewer’s comment, we have indicated that the previous work was published in 2015.

Line 213: why is the pepper HSP40 family with 76 annotated proteins larger than the Leishmania one with 77 proteins?

Unfortunately, for the Leishmania’s ego, the HSP40 family in pepper is larger than in Leishmania. This parasite has 72 distinct HSP40s, although there is a protein named J77 (Table 1). The reason for this discordance is due to the fact that HSP40 nomenclature was created from a study dealing with the characterization of this gene family in the order Trypanosomatida, using the combined information derived from the genomes of Trypanosoma cruzi, Trypanosoma brucei and L. major (Ref. 31). Altogether, 74 different HSP40s were identified, but in the L. major genome were found 69 HSP40s. Thus, Leishmania lacks some HSP40s present in Trypanosoma and the converse is also true. Now, in this study, we have identified three additional HSP40-genes that were not identified previously (neither in the Trypanosoma species) and they were named J75, J76 and J77. In sum, today, we can conclude that the number of HSP40s in Leishmania is 72.  

There are 8 type I, 20 type II, 38 type III, and 7 type IV HSP40/J proteins. Is there any significance to this distribution?

This distribution is based on structural features but, to the best of our knowledge, there are no clues regarding whether or not this classification has a functional correlation. Reviewer-2 alerted us about a recent work in which some type II HSP40s of humans exhibited an autoinhibitory mechanism in their interaction with HSP70, but more studies are needed to establish whether this mechanism is inherent to every type II HSP40s from every origin.

Although the manuscript is generally well written, there are a few awkward or grammatical incorrect sentences:

Line 39: “These proteins, many of them also known as 39 heat shock proteins (HSPs) because their involving in protecting cells…” should be “…because they are involved…”

Line 55: “…the latter often resulting fatal if untreated” should be “…the latter often resulting fatal if untreated

Line 62: “When the sand fly vector bites for feeding to the mammalian host…” should say something like “When the sand fly vector bites to feed on the mammalian host”.

Line 75: ” …because their stressful life cycles…” should be “…because of their stressful life cycles…”

Line 90: “members of HSP70 family” should be “members of the HSP70 family”.

Line 109: “…steps are important to stabilize non-native proteins and promoting their correct folding.” should be “steps are important to stabilize non-native proteins and to promote their correct folding.”

Line 30: “…whose prototype is that defined in Escherichia coli DnaJ” should be “…whose prototype is that defined in the Escherichia coli DnaJ protein”.

Line 133: “Remarkably, but no surprisingly…” should be “Remarkably, but not surprisingly”.
Line 144: “It is belief that J-domain…” should be “It is believed that the J-domain…”

Thank you for alerting us to these grammatical errors. They have been corrected in the revised manuscript.

Reviewer 2 Report

Molecular chaperons are highly versatile group of proteins which are key players of protein homeostasis network. The present review article written by Solana, CJ and co-authors presents a balanced view of HSP40 family of proteins from genus Leishmania, shedding light on their structural features, sequence, and possible functional diversities. Chaperons have a greater significance in the lifecycle of Leishmania parasites especially in the life form inside mammalian host and thus holds clinical relevance too. Nοnethless, this class of proteins remains largely understudied in this organism and awaits attention.  

The review is well written, informative and evokes readers interest in the topic. I am very positive about publishing this article in the journal’s special issue ‘The stress response in Microorganisms’. However, I have following suggestions to improve the overall quality of the article .

  1. Figure 1 which shows the overview of HSP70/40 chaperone system as a conceptual figure of the paper lacks clarity. The authors should elaborate figure by adding more information. For example, including the HSP70/40 interacting stage in the figure will help to make it more comprehensive and emphasize the role of HSP40.

  1. I assume that the identification of most of HSP40 family members is based on the homology studies conducted by the authors. In that case how confident are they on their predictions. It will be helpful to add another column to the tables in main text and supplementary files which shows the sequence similarity/identity of the Leishmanial orthologs to their closest higher eukaryotic counterparts.

  1. The authors seemed to have missed to refer some important articles which are relevant to the topic discussed – For example

 Faust, O., Abayev-Avraham, M., Wentink, A.S. et al. HSP40 proteins use class-specific regulation to drive HSP70 functional diversity. Nature 587, 489–494 (2020)

  1. Gu, Z. Liu, S. Zhang, Y. Li, W. Xia, C. Wang, H. Xiang, Z. Liu, L. Tan, Y. Fang, et al.

Hsp40 proteins phase separate to chaperone the assembly and maintenance of membraneless organelles

Proc. Natl. Acad. Sci. U.S.A., 117 (2020), pp. 31123-31133

Qiu XB, Shao YM, Miao S, Wang L. The diversity of the DnaJ/Hsp40 family, the crucial partners for Hsp70 chaperones. Cell Mol Life Sci. 2006 Nov;63(22):2560-70

I will recommend referring these articles in appropriate context in the text.

Minor comment.

There are typos in the text. The article will benefit from further proofreading.

Author Response

Molecular chaperons are highly versatile group of proteins which are key players of protein homeostasis network. The present review article written by Solana, CJ and co-authors presents a balanced view of HSP40 family of proteins from genus Leishmania, shedding light on their structural features, sequence, and possible functional diversities. Chaperons have a greater significance in the lifecycle of Leishmania parasites especially in the life form inside mammalian host and thus holds clinical relevance too. Nοnethless, this class of proteins remains largely understudied in this organism and awaits attention.  

The review is well written, informative and evokes readers interest in the topic. I am very positive about publishing this article in the journal’s special issue ‘The stress response in Microorganisms’. However, I have following suggestions to improve the overall quality of the article.

We are glad of reading this general comment on the manuscript, and we are grateful for the reviewer`s comments and suggestions.

  1. Figure 1 which shows the overview of HSP70/40 chaperone system as a conceptual figure of the paper lacks clarity. The authors should elaborate figure by adding more information. For example, including the HSP70/40 interacting stage in the figure will help to make it more comprehensive and emphasize the role of HSP40.

 Following this suggestion, we have elaborate further on the figure, including an additional stage in which the HSP70/40 interaction is pictured. With this modification, we believe that the critical roles played by HSP40 in the HSP70 chaperoning cycle are shown in a manner more explicit. Also, we have labelled when HSP70 is bound to either ATP or ADP, which is another relevant feature in the HSP70 cycle.

  1. I assume that the identification of most of HSP40 family members is based on the homology studies conducted by the authors. In that case how confident are they on their predictions. It will be helpful to add another column to the tables in main text and supplementary files which shows the sequence similarity/identity of the Leishmanial orthologs to their closest higher eukaryotic counterparts.

Following this suggestion (and suggestions by reviewer-1), we have created a new table (Table 2, which was incorporated in the main text) in which the orthologous proteins are listed. In that table, the sequence identity values have been added.

  1. The authors seemed to have missed to refer some important articles which are relevant to the topic discussed – For example

 - Faust, O., Abayev-Avraham, M., Wentink, A.S. et al. HSP40 proteins use class-specific regulation to drive HSP70 functional diversity. Nature 587, 489–494 (2020) [Ref. 15]

- Gu, Z. Liu, S. Zhang, Y. Li, W. Xia, C. Wang, H. Xiang, Z. Liu, L. Tan, Y. Fang, et al. Hsp40 proteins phase separate to chaperone the assembly and maintenance of membraneless organelles Proc. Natl. Acad. Sci. U.S.A., 117 (2020), pp. 31123-31133 [Ref. 78]

 - Qiu XB, Shao YM, Miao S, Wang L. The diversity of the DnaJ/Hsp40 family, the crucial partners for Hsp70 chaperones. Cell Mol Life Sci. 2006 Nov;63(22):2560-70 [Ref. 16]

 I will recommend referring these articles in appropriate context in the text.

 We agree that these are three relevant articles to the topic, and thanks for bringing them to our attention. These have been included in the revised manuscript (the reference number in the manuscript is indicated above).

Minor comment.

 There are typos in the text. The article will benefit from further proofreading.

We have carefully revised the text in order to catch typos, hoping to have removed all of them.
